# Plant-Associated *Bacillus thuringiensis* and *Bacillus cereus*: Inside Agents for Biocontrol and Genetic Recombination in Phytomicrobiome

**DOI:** 10.3390/plants12234037

**Published:** 2023-11-30

**Authors:** Antonina Sorokan, Venera Gabdrakhmanova, Zilya Kuramshina, Ramil Khairullin, Igor Maksimov

**Affiliations:** 1Institute of Biochemistry and Genetics of the Ufa Federal Research Centre of the Russian Academy of Sciences, 450054 Ufa, Russia; fourtyanns@googlemail.com (A.S.); gabdrakhmanovavenera@gmail.com (V.G.); krm62@mail.ru (R.K.); 2Sterlitamak Branch of Federal State Budgetary Educational Institution of Higher Education “Ufa University of Science and Technology”, 49 Lenin Avenue, 453103 Sterlitamak, Russia; z.m.kuramshina@strbsu.ru

**Keywords:** endophytes, insecticide, nematicide, Cry, Vip, *Bt*-crops, RNA-interference, plant productivity

## Abstract

*Bacillus thuringiensis* Berliner (*Bt*) and *B. cereus* sensu stricto Frankland and Frankland are closely related species of aerobic, spore-forming bacteria included in the *B. cereus* sensu lato group. This group is one of the most studied, but it remains also the most mysterious species of bacteria. Despite more than a century of research on the features of these ubiquitous bacteria, there are a lot of questionable issues related to their taxonomy, resistance to external influences, endophytic existence, their place in multidimensional relationships in the ecosystem, and many others. The review summarizes current data on the mutualistic relationships of *Bt* and *B. cereus* bacteria with plants, the structure of the phytomicrobiomes including *Bt* and *B. cereus*, and the abilities of plant-associated and endophytic strains to improve plant resistance to various environmental factors and its productivity. Key findings on the possibility of the use of Cry gene promoter for transcription of the target dsRNA and simultaneous release of pore-forming proteins and provocation of RNA-interference in pest organisms allow us to consider this group of microorganisms as unique tools of genetic engineering and biological control. This will open the prospects for the development and direct change of plant microbiomes, and possibly serve as the basis for the regulation of the entire agroecosystem.

## 1. Introduction

Currently, in *B. cereus*-group bacteria, *Bacillus thuringiensis* Berliner (*Bt*) is one of the most studied microorganisms—one of the Alpha and Omega of the modern biological control strategy. *Bt*-based pesticides account for up to 75% of the global bioinsecticide sales market and about 4% of all insecticides [1]. It has been continuously used in agriculture and forestry for more than 50 years [1,2,3]. Rod-shaped, poorly motile, spore-forming, facultatively anaerobic, synthesizing insectotoxic proteins bacteria *Bt* (Bacteria; Terrabacteria group; Bacillota; Bacilli; Bacillales; Bacillaceae; *Bacillus*; *Bacillus cereus* group) subspecies *thuringiensis*, *kurstaki*, *aizawai*, *tenebrionis*, and *israelensis* are most often used as the basis of bioinsecticides worldwide [1,4]. In the past decade, research on the development and use of biological control agents based on *Bt* in vector control programs has been greatly stimulated for the prevention of dangerous human diseases spreading [1,2,3]. Due to their high efficiency against various pests and environmental safety, many researchers consider *Bt*-based agents to be an effective and environmentally friendly alternative to chemicals [1,2,3,4,5]. Unfortunately, the use of *Bt* raises a number of problems, which will be discussed below.

*Bt* insecticidal proteins possess susceptibility to a broad spectrum of environmental influences, among which ultraviolet (UV) rays are the most important [6,7]. Ultraviolet radiation causes shorter persistence of *Bt* under the field conditions [8] due to extensive bacterial DNA cleavage, in particular, double-strand breaks [9]. Unfortunately, it demands the repeated spraying of crops, and the cost of using *Bt* is rising. Therefore, creation of light-stable biocontrol agents based on *Bt* strains with higher insecticidal efficiency is of interest.

*Bt* is accepted by many authors as a single species of *B. cereus* group, but from different points of view, *Bt* can be seen as an independent species or a subspecies of *B. cereus* sensu lato, bearing plasmids encoding insectotoxic proteins [10,11]. According to several experts, *Bt* are insectotoxin producers and belong to the supraspecific group of *B. cereus* sensu lato, which includes 21 closely related species, including *Bt*, *B. mycoides* Flügge 1886; *B. weihenstephanensis* Lechner et al. 1998; *B. pseudomycoides* Nakamura 1998; *B. anthracis* Cohn 1872: *B. cereus* sensu stricto; *B. gaemokensis* Jung et al. 2010; *B. manliponensis*; *B. toyonensis*; *B. bingmayongensis*; *B. cytotoxicus*; and *B. wiedmannii* [12]. *B. cereus* sensu stricto (Bacteria; Terrabacteria group; Bacillota; Bacilli; Bacillales; Bacillaceae; *Bacillus*; *Bacillus cereus* group) toxins can cause gastrointestinal diseases, wound or systemic infections, and eye infections [11]. Perhaps because of the controversies surrounding the approach, a lot of molecular investigations have shown that strains of these species are not distinguishable, since DNA–DNA hybridization similarities and the average nucleotide identity between these *Bacillus* types are slightly higher than the levels used to distinguish between closely related species, but *B. cereus* sensu stricto and Bt should continue to be recognized as validly published species [13,14]. The extensive analysis of genomic data show that the distribution of insecticidal genes is irregular and numerous strains identified as Bt can be assigned to polyphyletic subclades within the *B. cereus/Bt* clade [14]. Thus, the presence of certain plasmid encoding Cry and Vip cannot be used as a diagnostic marker of *Bt*. According to the composition vector tree (CVTree) method, there is the relationship between Bt and other *B. cereus* sensu lato species, but it was the best option to be used for typing Bt strains [15]. On the basis of average nucleotide identity researchers set up *Bt* (so-called *B. cereus* sensu stricto serovar Berliner biovar Thuringiensis by authors) in *B. cereus* sensu stricto genomospecies containing 949 genomes [16]. The proximity of *Bt* to pathogenic microorganisms, for example to *B. cereus*, put forward the problem of their identification and separation in food products [17,18]. Now, *Bt* residues, which cannot be distinguished from natural populations of *B. cereus* in routine food safety diagnostics are enumerated as “presumptive *B. cereus*” [17], and approaches to its differentiation are often based on Cry-genes presence [18]. Thus, the information on *Bt* and *B. cereus* features, set out in this review, is based on the definition of strains and isolates proposed by researchers, but it should be noted that some of them can be attributed to each other. Analysis of the above sources allows us to conclude that it is necessary to draw attention to the problem with the identification of *B. cereus* group using various taxonomic methods.

One of the important problems of the prolonged and large-scale pest control using *Bt* (*Bt*—crops) is the appearance of tolerant or resistant to *Bt* insect populations [19,20]. A high level of pest resistance to *Bt*-toxins is based on the mutations/reduction in receptors to insectotoxins on the midgut epithelium [21], the development of the humoral responses, including the synthesis of antimicrobial peptides, the activation of the polyphenol oxidase system, the generation of reactive oxygen species, encapsulation of crystals and activation of phagocytosis systems [22].

Previously, *Bt* strains were considered a cosmopolitan soil bacteria with occasional insecticidal activity [23]. Now, the niche of these bacteria is more often attributed to the phylloplane, considering them to be mutualists in relation to plants [24], and in our opinion, it is a kind of revolution in the concept of ecosystems. Plants intimately interact with diverse communities of microorganisms, such as bacteria, fungi, nematodes, protists, and viruses that colonize all plant tissues, rhizosphere, and soil [1,19,23,25]. The microbiome establishes complex and dynamic interactions with the host plant and can improve plant resilience to environmental stresses due to the high level of flexibility of these important genetic resources of the whole plant/microbiome system [25,26].

According to modern data, among the bacterial species isolated from internal plant sites, the frequency of the detection of bacteria of *B. cereus* group, along with *B. megaterium*, is the highest [25,26,27,28]. Endophytic *Bt* have been isolated from plants of the dicotyledonous and monocotyledonous families, from ferns and bryophytes, distributed in all hemispheres (Table 1). However, *Bt* strains, which were previously identified as endophytic, may have quantitative differences in population density in tissues of wheat plants [29] and in various organs (roots and shoots) of potato plants [30]. Possibly, the endophytic lifestyle of *Bt* is one of the solutions of their UV susceptibility [6,7]. At these time, it should be remembered that *Bt* strains can actively produce some exotoxins that are detrimental to eukaryotes: α-exotoxin (phospholipase C); β-exotoxin (thuringeinsin, toxic to mammalian); γ-exotoxin (toxic to sawfly insects); lice death factor; and exotoxin mouse death factor (toxic to mice and lepidoptera) [31]. In particular, β-exotoxins occurrence limits the application of some *Bt* strains [32], and the endophyticity of *Bt* may also represent a hypothetical problem since *Bt* cells in this case are not removed from the plant surface and can be eaten.

The use of *Bt*-based biocontrol agents, as well as *Bt*-crops, can change the maintenance of the microbial population in the endo-, rhizo-, and phyllosphere of plants, directly influencing microorganisms, in particular, nitrogen-fixing bacteria, or modifying plant immune status [24,33,34]. It was found that endophytic *Bt* strains are able to colonize the nodule roots of *Erythrina brucei* Schweinf. emend. Gillett [35]; *Glycine max* L.; *Vigna umbellata* Thunb.; *Phaseolus vulgaris* L. [36]; *Zea maize* L. [37]; and *Macrotyloma uniflorum* Lam. [38]. de Alameida et al. [37] showed the possibility of increasing the growth-promoting effect of *Azospirillum brasilense* Tarrand, Krieg & Döbereiner, 1978 Ab-V5 in combination with the application of endophytic *Bt* RZ2MS9 labeled with green fluorescent protein. Thus, the possibility of co-inoculation of maize with phosphate-mobilizing *Bt* B116 and nitrogen-fixing *Azospirillum* sp. (strains A1626 and A2142) was demonstrated [39].

Delanthabettu et al. [40] identified twelve *Bt* strains, producing crystal structures of Cry proteins, from cowpea root nodules. PCR analysis of those strains revealed the occurrence of Cry-genes of different families. However, endophytic *Bt* KMCL07 showed quorum quenching activity, and AiiA lactonase KMMI17 production by this strain inhibits biofilm formation and attenuates the pyocyanin of Gram-negative bacteria *Pseudomonas aeruginosa* PAO1 [41]. It is possible that chitinase produced by *Bt* subsp. *pakistani* HD 395 can destroy the Nod factor of the soybean symbiont *Bradyrhizobium japonicum* Kirchner 1896, preventing root noduation [42]. Since *Bt* strains exhibit activity against pathogenic fungi [43], it can be possible that arbuscular mycorrhiza may be interfered by *Bt* strains [33].

Thus, searching for biocontrol agents is based on the finding of bacteria that have a multidimensional effect on pest organisms and increase the development time of their resistance. Now it is clear that the paradigm of the use of *Bt* in the same way as chemical pesticides is ineffective, and that it is necessary to investigate *Bt* as a part of the microbiome of the whole biocenosis.

**Table 1 plants-12-04037-t001:** *B. thuringiensis* and *B. cereus* endophytic strains and their properties.

Strain/Isolate	Plant	Properties	Pathogens and Pests Susceptible to Strain	Ref.
*Bt* subsp. *kurstaki* HD-1 (*Bt*k); S1450; *Bt* S1905; *Bt* S2122; *Bt* S2124	*Brassica oleracea* var. *Capitate* L.	Pest control	*Plutella xylostella* L.	[25]
*Bt* B-5689	*T. aestivum* L.	Control of insect and fungi, triggering ISR	*Stagonospora nodorum* (Berk.) Castellani & E.G. Germano, *Shizaphis graminum* Rondani	[29]
*Bt* B-6066	*S. tuberosum* L.	Control of insect and oomycetes, triggering ISR	*Leptinotarsa decemlineata* Say, *Phytophthora infestans* Mont. (de Bary)	[30]
*Bt* VLS72.2; *Bt* VLS64.1; *Bt* VLS64.3; *Bt* VLS21; *Bt* VRB1; *Bt* VLG15; *Bt* VL4b; *Bt* VL4C; *Bt* VL2d; *Bt* VL126	*Glycine max* (L.) Merrill, *Vigna umbellata* (Thunb.) Ohwi & H. Ohashi, *Macrotyloma uniflorum* (Lam.) Verdc.; *Lens culinaris* Medik.	Pest control	*Spilosoma obliqua* Walker	[36]
*Bt* NEB17	*G. max* (L.) Merrill	Nodulation improvement, plant growth promotion, and yield increase	N/A	[39,40]
*Bt* KMCL07	*Pueraria thunbergiana* Parl.	Inhibition of pathogen growth	*Pseudomonas aeruginosa* (Schroeter) Migula	[41]
*Bt* subsp. *kurstaki* HD-1	*Gossypium hirsutum* L.	Pest control	*Spodoptera frugiperda* J. E. Smith, *Plutella xylostella* L.	[44]
*Bt* 2810-S-6; *Bt* 65-S-35; *Bt* 2810-S-4	*Trifolium hybridum* L.	Pest control	*Pieris brassicae* L.	[45]
*Bt* S1450; *Bt* S1302; *Bt* S1989	*Citrus sinensis* (L.) Osbeck	Pest control	*Diaphorina citri* Kuwayama	[46]
*Bt israelensis* LBIT-1250L *Bt kurstaki* LBIT-1251P	*Lavandula angustifolia* Mill.; *Euphorbia pulcherrima* Willd. ex Klotzsch	Pest control	*Aedes aegypti* L.; *Manduca sexta* L.	[47]
*Bt AK08*	*Theobroma cacao* L.	Nematicidal activity	*Meloidogyne incognita* Kofoid & White	[48]
*Bt* KL1	*Andrographis paniculata* Nees.	Probiotic, antimicrobial activity	*Vibrio parahaemolyticus* Sakazaki et al.; *Aeromonas caviae* Popoff, *Proteus vulgaris* Hauser	[49]
*Bt* GS1	*Pteridium aquilinum* (L.) Kuhn	Antimicrobial activity	*Rhizoctonia solani* J.G. Kühn	[50]
*Bt* isolates 5; 6; 7; 8; 9; 10; 11; 12; 13; 26; 27	*Chelidonium majus* L.	Antimicrobial activity	*A. alternata* (Fr.) Keissl.; *Chaetomium* sp.; *Paecilomyces variotii* Bainier, *Exophiala mesophila* Listemann et Freiesleben	[51]
*Bt* EB 69	*Physalis alkekengi* L.	Antimicrobial activity	*Staphylococcus aureus* Rosenbach, *Citrobacter freundii* Werkman and Gillen, *Proteus mirabilis* Hauser, *Shigella flexneri* Castellani & Chalmers	[52]
*Bt FVA 2–3*	*Capsicum annuum* L.	Antimicrobial activity, triggering ISR	*Botrytis cinerea* Pers.	[53]
*Bt H1R2*	*S. lycopersicum* L.	Antimicrobial activity, growth promotion	*B. cinerea* Pers.	[54]
*Bt* 58-2-1, *Bt* 37-1	*T. aestivum* L.	Antimicrobial activity, yield increase	*Urocystis tritici* Koern.	[55]
*Bt* C3	*Manihot esculenta* Crantz	Antimicrobial activity	*Aspergillus flavus* Link, *A. niger* van Tieghem	[56]
*Bt* TbL-22 *Bt* TbL-26	*Taxus brevifolia* Nutt.	Antimicrobial activity	*Xanthomonas citri* subsp. *citri* (ex Hasse) Gabriel et al.	[57]
*Bt* SBL3	*Berberis lycium* Royle	Antimicrobial activity	*Listeria monocytogenes* (E. Murray et al.) Pirie, *Escherichia coli* (Migula) Castellani and Chalmers, *A. niger* van Tieghem, *A. flavus* Link	[58]
*Bt* B-56	*Withania somnifera* (L.) Dunal	IAA production	N/A	[59]
*Bt* RZ2MS9	*Paullinia cupana* Kunth	IAA production, phosphate solubilization, nitrogen fixation, metal chelation	N/A	[60]
*Bt* AZP2	*Pinus ponderosa*	Increase drought toleration	N/A	[61]
*Bt* Fse6 *Bt* Fse8	*Oryza rufipogon* Griff.	Siderophores and IAA production, phosphate solubilization	N/A	[62]
*Bt* Y2B	*Cicer arietinum* L.	Siderophores, hydrogen cyanide and IAA production, phosphate solubilization	N/A	[63]
*Bt* W65	*S. tuberosum* L.	Increase in potato yield	N/A	[64]
*Bt* BMG1.7; *Bt* HD22; *Bt* HD868; *Bt* H77; *Bt* H112; *Bt* H156; *Bt* H172	*T. aestivum* L.	Auxin production, ethylene balance control	N/A	[65]
*B. cereus* GX1	*Garcinia xanthochymus* Hook.f. ex T.Anderson	Antimicrobial activity	*Pseudomonas aeruginosa* (Schroeter) Migula, *E. coli* (Migula) Castellani and Chalmers, *Salmonella typhi*, *Staphylococcus aureus* Rosenbach	[66]
*B. cereus* BCM2	*Fragaria ananassa* (Duchesne ex Weston) Duchesne ex Rozier	Growth regulation, nematocidal activity	*Meloidogyne incognita* Kofoid & White	[67]
*B. cereus* YN917	*O. sativa* L.	Antimicrobial activity, plant growth-promoting activity, production of IAA, siderophores, ACC deaminases, proteases, β-1,3-glucanase, amylases, cellulases,	*Magnaporthe oryzae* (T.T. Hebert) M.E. Barr	[68]
*B. cereus* XB177R	*S. melongena* L.	Antimicrobial activity, destruction of fungal cell walls (chitinases)	*Ralstonia solanacearum* (Smith) Yabuuchi et al.	[69]
*B. cereus* LBL6	*Berberis lycium* Royle	Antimicrobial activity	*P. aeruginosa* (Schroeter) Migula, *Bacillus spizizenii* (Nakamura et al.) Dunlap et al.; *Salmonella typhimurium* *Acinetobacter baumannii* Bouvet and Grimont *A. niger* van Tieghem, *A. flavus* Link	[58]
*Bt* GDB-1	*Alnus firma* Siebold & Zucc.	Growth regulation, phytoremediation	N/A	[70]
*Bt PB2*	*V. faba* L.	Growth regulation auxins and ammonia production	N/A	[71]
*B. cereus* AKAD A1-1	*Glycine max* (L.) Merr.	Growth regulation, ACC deaminase production	N/A	[72]
*B. cereus* KP120	*Kosteletzkya virginica* (L.) C. Presl ex A. Gray	Increase in halotolerance	N/A	[73]
*B. cereus* HK012	*Kosteletzkya virginica* (L.) C. Presl ex A. Gray	ACC deaminase production, increase in salt stress toleration	N/A	[74]
*B. cereus* C1L	*Zea mays* L.	Triggering systemic resistance	N/A	[75]
*B. cereus* S2	*Camellia sinensis* (L.) Kuntze	Ammonia, cellulase and protease production	N/A	[76]
*B. cereus*, *Bt*	*Pennisétum gláucum* (L.) R.Br.	Drought tolerance	N/A	[77]
*Bt* and *B. cereus*	*Pinellia ternata* (Thunb.) Makino	Purine alkaloids guanosine and inosine production	N/A	[78]
*B. cereus*	*B. napus* L.	Antioxidative activity, production of N6-(Δ2-isopentenyl) adenine, IAA, and siderophores	N/A	[28]
*B. cereus* AV-12	*V. mungo* (L.) Hepper	ACC deaminase production, drought tolerance biofertilizer in fields affected, Ba, and Ni	N/A	[79]
*B. cereus* LN714048	*Cenchrus ciliaris* L.	Increase in salt tolerance and yield of wheat	N/A	[80]
*B. cereus* SA1	*Echinochloa crus-galli* (L.) Beauv.	Production of gibberellin, indole-3-acetic acid, and organic acids, increase drought tolerance in plants	N/A	[81]
*B. cereus* T4S	*Helianthus annuum* L.	Growth promotion	N/A	[82]

## 2. Spectrum of *Bt* and *B. cereus* Availabilities

Now, it is believed that systems of plants and their associated microbiota resulting from the evolutionary selection contributes to the overall stability of the whole holobiont [23,33,83]. The biocontrol of pathogens and pests by benefit microorganisms, originating from the competition for niches/nutrients, is possibly one of the most important, at least now, features of microbiomes in agrocenosis [23,24]. Mechanisms of biocontrol on direct (the production of antimicrobial/insecticide compounds) and indirect (the induction of systemic resistance in plants) means can be conditionally divided [33]. These branches determine all spectra of biocontrol possibilities of plant-associated microbes, including, for example, pathogen quorum sensing interference and the altering of the soil microbiota [83].

### 2.1. Production of Insectotoxic Proteins

The insecticidal activity of *Bt* is based on their ability to synthesize toxic Cry (and any other) proteins that cause the death of more than 3000 insect species from 16 orders [4]. More than 800 sequences of genes encoding insecticidal Cry proteins have now been identified in the plasmid genome of various *Bt* strains [5]. The products of these genes usually accumulate in the form of crystalline inclusions in bacterial cell compartments, which can account for 20 to 30% of the dry weight of sporulating cells [1,2,3,4,5]. Cry proteins possess the selective insectotoxicity in relation to insects from families Lepidoptera, Diptera, Coleoptera, Rhabditida, Hemiptera, Hymenoptera, Gastropoda. CryI proteins, for example, showed toxicity to Lepidopteran insects, CryII to Lepidoptera and Diptera, CryIII to Coleoptera, CryIV to Diptera exclusively, and CryV to Coleoptera and Lepidoptera [1,4,84,85]. Recently, Cry1Ab and Cry1Ac proteins are found to be toxic to cervical cancer (HeLa) cells were also found [86].

*Bt* also secrete Vip (vegetative insecticidal protein) and Sip (secreted insecticidal protein) toxins during the vegetative phase of culture growth [4,83]. Toxins Vip1 and Vip2 have high insecticidal activity against coleopteran and hemipteran pests, Vip3—against Lepidoptera [83]. The insecticidal activity of phylogenetically close to Vip1 family Vip4 proteins has not been understood in detail. Sip proteins show an insecticidal effect on beetle larvae [87]. The activity of Cry proteins against Hemiptera is often little due to suboptimal conditions for the activation, processing, and binding of Cry and Vip proteins in the hemipteran gut [2]. Among these, aphidicidal Cry-related proteins include Cry73Ba1/Cry73Ba2 (against *Myzus persicae* Sulzer, 1776) [87], Cry51Aa2 (against *Lygus hesperus* Knight, 1917) [88], and Cry64Ba-Cry64Ca (against *Laodelphax striatellus* Fallén, 1826 and *Sogatella furcifera* Horváth, 1899, respectively) [89]. The *Bt* BST-122 spore and crystal mixture show toxicity to coleopterans and two-spotted spider mite *Tetranychus urticae* Koch, 1836 due to a novel Cry5-like protein production [90]. It was found that toxic Cry5B proteins can play an important place in the anthelmintic activity of *Bt* bacteria [91]. However, *Bt* KAU 50-producing Cry6, Cry16, Cry20 and *Bt* KAU 424-producing Cry1and Cry14 show nematocidal activity against *Haemonchus contortus* (Rudolphi, 1803) Cobb, 1898 as well [92].

Of particular interest is the fact that the production of Cry proteins in *Bt* cells is under the regulatory control of bacterial miRNAs, depending on its interaction with a potential host. For example, *Bt* strain Y*Bt*-1518 regulates the accumulation of the Cry5Ba protein only in the infected nematode organism with the help of microRNA [93]. The suppression of Cry5Ba synthesis is due to the cyclic *Bt*sR1 RNA binding to the gene *Cry5Ba* transcript through direct base pairing. It has been reported that strains *B. cereus* BCM2 and *B. cereus* SZ5, which were described as endophytes of strawberries (*Fragaria ananassa* (Duchesne ex Weston) Duchesne ex Rozier, (1785)) and eastern persimmon (*Diospyros kaki* Thunb.) exhibit high nematocidal activity against *Meloidogyne incognita* Kofoid & White, 1919 on tomato plants [67]. The impact of *Bt* on nematodes *M. hapla* Chitwood, 1949, which worsens their reproductive properties and efficiency of tomato root colonization, is connected with the synthesis of the Cry6A insectotoxin [94]. *Bt* BRC-XQ12 produces Cry1Ea11 protein, which is effective against the coniferous nematode *Bursaphelenchus xylophilus* Steiner & Buhrer, 1934 [95]. It should be noted that representatives of other types of bacteria, such as *Clostridium bifermentans* Weinberg and Séguin 1918; *Paenibacillus popiliae* Dutky 1941; *Paenibacillus lentimorbus* Dutky 1940 (Lists 1980); and *Bacillus sphaericus* Meyer and Neide 1904, are also capable of accumulating toxins like Cry and Cyt [10,31]. *Lysinibacillus sphaericus* (Meyer and Neide 1904; *Paenibacillus alvei* Cheshire; and Cheyne 1885 can produce sphericolysin/anthrolysin, *Clostridia* sp. and *Aeromonas* sp.—hydrophilaprotein Mtx2 and aerolysin (aerolysin), respectively. Various *L. sphaericus* strains also produce BinA/B toxins, Mtx 1-4, spherolysin, Cry48, and Cry49, and *Photorhabdus* sp. can produce PirA/B and Mcf toxins [10]. These genes can be of value to the development of genetically improved insecticidal strains of *Bt*.

### 2.2. Production of Various Classes of Compounds, Apart from Cry Proteins

#### 2.2.1. *Bt* against Insect Vectors of Viruses and Its Potential Direct Influence on Viral Particles

It is well-known that aphids (Aphidoidea), whiteflies (Aleyrodidae), thrips (Thysanoptera), nematodes of the genera *Trichodorus*, and *Paratrichodorus* intensively transfer many viruses from plant to plant [1,2,29,96]. Now, the application of insecticides to control virus vectors is almost the only reliable way to limit viruses from spreading [2,97]. Cry proteins of *Bt* possess limited field and laboratory efficacy against Hemiptera [2]. For example, a Cry41-related toxin had moderate toxic activity against *M. persicae*, and this effect was achieved, among other things, due to the influence of aphid endosymbiont *Buchnera* sp. [98]. Pests of this group, in contrast to chewing pests (Lepidoptera and Coleoptera), have piercing-sucking oral apparatus and feed on the phloem (whiteflies, aphids, and mealybugs), xylem (spittlebugs and sharpshooters), or juice of seeds and fruit (stinkbugs) and do not eat cells or Cry crystals on the surface of plants. The prevalence of *Bacillus* spp. in plant sap, in our opinion, can be improved using endophytic strains of *Bt*. Endophytic *Bt* B-5351 as they killed about 60% greenbug aphids and stopped their reproduction on isolated leaves of wheat, which were immersed in *Bt* B-5351 suspension, and increased transcriptional activity of potato PR-genes [29]. Under the field conditions, this *Bt* strain reduced the severity of PVY, PVM, and PVS on potato plants [30]. *Bt* strains produce other classes of compounds that can be toxic for hemipteran pests or have direct antiviral activity, in particular, lipopeptides iturin, surfactin, and fengycin [29,99].

Extracellular ribonucleases (RNases), which are produced by *B. amyloliquefaciens*; *B. pumilus*; *B. licheniformis* [100,101]; and *Bt* [102] can hypothetically be used for protecting plants and other organisms from viruses [99]. It was reported that preparations based on *Bt* culture fluid with the maximum yield of secreted RNases were effective against A/Aichi/2/68 (H3N2) human influenza virus in experiments on infected mice, close to that of the reference drug Tamiflu [102].

#### 2.2.2. Production of Nematocidal Compounds

Plant-damaging nematodes have a high-level resistance to Cry toxins. Thus, the resistance to Cry1Ac in *Trichoplusia ni* Hübner is due to multi-gene mutations in ABCC2 gene and the altered expression of APN1 and APN6 genes [103]. Therefore, it is even suggested to study the effect of other *Bt* metabolites on nematodes. The treatment of tomato plants with endophytic strain *Bt* AK08 resulted in 95.46% mortality of nematodes *Meloidogyne* sp., which, as the authors believe, is associated with the production of nematocidal cholest-5-en-3-ol(3.beta.)-carbonochloridate [48]. High nematocidal activity against juvenile *M. incognita* parasitizing on tomato roots was demonstrated by the endophytic strain *B. cereus* BCM2 The authors of the work believe that this effect of the strain is associated with the production of 2,4-di-tert-butylphenol and 3,3-dimethyloctane [67]. *Bt* GBAC46 and *Bt* NMTD81 strains isolated from plants of the Qinghai–Tibetan Plateau possess high nematocidal activity due to properties of the bacteria themselves and their ability to induce defense reactions of *Oryza sativa* plants against the nematode *Aphelenchoides besseyi* [104].

The treatment of grape plants with a multicomponent mixture of *Bt* FS213P; *Bt* FB833T; *B. amyloliquefaciens* FR203A; *B. megaterium* FB133M; *B. weihenstephanensis* FB25M; *B. frigoritolerans* FB37BR; and *P. fluorescens* FP805PU strains showed effectiveness against *Xiphinema* sp. and *Meloidogyne* sp. nematodes comparable with the action of a chemical nematicide [105]. In our opinion, the last mentioned data are very important since it demonstrates the possibility of the artificial construction of plant microbiomes, thereby changing plant phenotype.

#### 2.2.3. Production of Fungicidal and Bactericidal Compounds and Triggering Systemic Resistance in Plants

During the last decade, the ability of *Bt* and *B. cereus* to control pathogenic components in agrocenoses has been widely taken into account [106,107]. It is evident that endophytic bacteria are natural competitors to invaders, and that their action combines antimicrobial and immunostimulating activities [refs below]. Wang M. et al. [108] showed that *Bt* 4F5 induced ISR through the jasmonic acid/ethylene (JA/ET) and salicylic acid pathways in *Brassica campestris* L. against the pathogen *Sclerotinia sclerotiorum* (Lib.) de Bary and the pest *P. xylostella*, engaging exopolysaccharides as elicitors. The protective role of the endophytic *Bt* FVA 2–3 associated with the roots of *Capsicum annuum* L. fruits against the fungus *B. cinerea*, which causes gray rot, is associated with the side with direct antagonism of the strain and with the induction of defense systems in plant cells [53]. While *Bt* serovar *aizawai* AbtS-1857, which is a part of the commercial bioinsecticide XenTari^®^, which was used against the same pathogen on tomatoes did not show direct antagonism against the fungus, it induced *PR-1* and *PR-5* genes transcription in plants, which determines the salicylate-dependent defense reactions [109]. In the same way, the protective effect of *B cereus* EC9 treatment against Fusarium wilt (*F. oxysporum* Schltdl., 1824) on tomato [110] and *Kalanchoe* sp. [111] was not associated with the direct antagonistic effect of bacteria on the fungus, but functioned indirectly, through the expression of plant genes associated with the JA signaling system. *B. cereus* AR156 treatment significantly suppressed the growth of gray mold on strawberry caused by *B. cinerea* and prevented senescence during storage. Treatment with *B. cereus* AR156 enhanced the reactive oxygen-scavenging and defense-related expression of salicylate-dependent PR-genes (PR1, PR2 (β-1,3-glucanases), and PR5 (thaumatin-like proteins)) [112]. Treatment with the same strain indirectly induced the systemic resistance of *A. thaliana* plants to *B. cinerea* through signaling defense pathways regulated by salicylic acid and mitogen-activating protein kinases [112], which are associated with the suppression of the accumulation of microRNA miR825/825*, as well as plant ubiquitin protein ligases (miR825) and TIR-NBS-LRR receptor proteins (miR825*) targeted against mRNA [113]. Transgenic *Arabidopsis* plants with the attenuated expression of miR825 and miR825* were more resistant to *B. cinerea* B1301, while plants overexpressing miR825 and miR825* were more susceptible to the pathogen as compared to wild-type plants. Accordingly, the transcription of defense-related genes and oxidative burst were faster and stronger in miR825 and miR825* knockdown plant lines [114]. In addition, it was shown that under the influence of the *B. cereus* AR156 strain, the accumulation of microRNA miR472, which negatively regulates the transcription of the NBS-LRR gene, was suppressed in *Arabidopsis* plants infected with *P. syringae* pv. *tomato* DC3000 [115]. Two transcription factors in *Arabidopsis* plants, WRKY11 (regulated by JA) and WRKY70 (regulated by salicylic acid), were identified as important regulators involved in induced systemic resistance which was observed under the influence of *B. cereus* AR156. Gene products modulated the *B. cereus* AR156-initiated defense cascade in an NPR1-dependent manner [115].

Plant-associated *Bt* and *B. cereus* strains are capable of producing a lot of volatile organic compounds (VOCs) [115], for example, some *Bt* strains from the rhizosphere of *Vigna subterranea* (L.) Verdc., 1980 [116]. Thus, *B. cereus* C1L produces dimethyl disulfide which induces systemic resistance in tobacco against *B. cinerea* [117]. The violation of glucose transport in the ptsG mutant line of the *B. cereus* C1L (the deficient synthesis of acetoin and 2,3-butanediol) led to the sygnificant decrease in its ability to exist endophytically in maize roots, as well as to the triggering of systemic resistance [75]. After whole genome sequencing, eight clusters of genes were identified in the genome of *B. cereus* D1 (BcD1), including those responsible for the synthesis of VOCs; serine proteases; plant-growth-stimulating metabolites, for example, indoleacetic (IAA); abscisic acid (ABA); and JA, were expressed under stress conditions [118]. The production of 3,5,5-trimethylhexanol, which disrupts the permeability of bacterial membranes by the *B. cereus* D13 strain, led to the growth inhibition of *R. solanacearum* and *P. syringae* pv. tomato DC3000 and *Xanthomonas oryzae* pv. *oryzicola* [119]. A strong inhibition of the growth of *S. sclerotiorum* mycelium (65.4%) under the influence of a VOC produced by *B. cereus*, CF4-51 (2-pentadecanone, bis (2-methylpropyl) ester of 6,10,14-trimethyl-1,2-benzenedicarboxylic acid, dibutyl phthalate), which is associated with changes in the expression of four genes of the pathogenic fungus, was associated with sclerotia formation (Ss-sl2, SOP1, SsAMS2, and SsSac1) [120].

The antifungal activity of *Bt* and *B. cereus* strains may be associated with their production of chitinases, glucanases, and proteases. Fuente-Sacido et al. [121] showed that the fungicidal activity of *Bt* subsp. *tenebrionis* DSM-2803 against the fungus *Colletotrichum gloeosporioides* (Penz.) Penz. & Sacc., 1884 responds to bacterial endochitinases. It was shown that oligosaccharides of chitin and chitosan produced with the participation of extracellular chitinases and chitosanases of *Bt* subsp. *dendrolimus* B-387 possess high bactericidal and fungicidal activity [122]. Similarly, endophytic *B. cereus* XB177R, which show high chitinase activity effectively protected eggplant plants from bacterial wilt caused by *R. solanacearum* [69].

*Bt* strains are known to produce antibiotic compounds that inhibit the growth of other bacteria, in particular, other *Bt* strains. Thus, it is known that *Bt* strains synthesize up to 18 different types of bacteriocins, for example, the turincin family, tochicin, turicin 7, entomocin 9, bacturicin F4, etc. [123]. *Bt* and *B. cereus* produce cyclic or linear heptapeptides kurstakins, which are able to damage the biological membranes of bacteria and fungi [124]. *Bt* NEB17 isolated from soybean nodules produced bacteriocin subclass IId (3.162 kDa) and turicin-17 with high antibacterial activity against rhizospheric pathogens [125]. The treatment of plants with turicine-17 resulted in the accumulation of proteins of the carbon and energy metabolism pathway, which, together with changes in the phytohormonal balance, compensate for the losses that occur during osmotic stress, besides its antimicrobial activity [126].

The information above suggests that the use of *Bt*-based biopreparations can alter the composition of the plant-associated microbial population, as well as in its environment, directly or indirectly through the influence on plants metabolism [33,68,81]. Considering these data, it can be concluded that there are endophytic polyfunctional *Bt* and *B. cereus* strains that possess not only insecticidal but also fungicidal, bactericidal, and viricidal properties against a broad spectrum of important pathogens, or have the ability to stimulate plant defense system, incidentally affecting the functioning of pathogen virulence factors.

#### 2.2.4. Improving of Plant Tolerance to Abiotic Stressors

Last decade it has been shown that some strains of *Bt* or *B. cereus* can protect plants from abiotic factors. Thus, the plant-associated strain *B. cereus* AKAD A1-1, which synthesizes ACC deaminase, increased the resistance of soybean to osmotic stress [72]. The treatment of rice plants growing in soils contaminated with arsenic with *Bt* contributed to a decrease in the accumulation rate of arsenic ions in grains and the tolerance of plants to toxic influence [127]. The formation of tolerance of black mustard plants to Cr^3+^ ions was found under the influence of *B. cereus* treatment due to ability of the strain under investigation to solubilize phosphate and synthesize siderophore, osmolyte (proline and sugar), and phytohormones in soils contaminated with chromium [128]. The cadmium-induced phytotoxic influence on pepper after the treatment of plants with *Bt* IAGS 199 and putrescine was decreased [129]. The exopolysaccharide of the *B. cereus* SZ-1 strain isolated from annual mugwort (*Artemisia annua* L., 1753) plants showed antioxidant activity against toxic 1,1-diphenyl-2-picrylhydracyl radicals [130]. The promotion of antioxidant activity has been shown in ryegrass plants inoculated with *Bt* EhS7 and *B. cereus* RA2 strains, which allowed plants to grow more efficiently in soils with a high content of heavy metals [131]. It was found that the influence of the halotolerant strain *Bt* PM25 on corn promotes plant growth in saline soils through the high antioxidant activity of bacterial metabolites [132]. ACC deaminase-producing plant-associated bacteria are of particular interest. This enzyme is therefore often designated as a “stress modulator”, limiting ethylene levels in plants [133,134]. Thus, endophytic *B. cereus* AV-12, isolated from *Vigna mungo* Hepper, 1956, synthesizes ACC deaminase [79]. *B. cereus* brm, which synthesizes ACC deaminase, significantly stimulated the growth of mung bean V. radiata plants and improved the viability of plants under salt stress [135]. There are some opinions that bacteria with high ACC-deaminase activity also have effective mechanisms of phosphate solubilization [136]. Interestingly, ACC deaminase-producing strain *Bt* GDB1 improved the efficiency of the phytoremediation of areas contaminated with heavy metals by *Alnus firma* Siebold & Zucc., 1846, a metal ion hyperaccumulator [70]. The insertion of the *acdS* gene encoding ACC deaminase from *B. cereus* HK012, an endophytic strain isolated from the root tuber of halophytic plant *Kosteletzkya virginica* L. in tobacco genome, contributed to an increase in biomass, chlorophyll content and, significantly, an up to 50% increase in proline content in plants exposed to salt stress (150 mmol/L^−1^ NaCl). Accordingly, tobacco plants expressing the *B. cereus* HK012 acdS gene showed higher salt tolerance than wild-type plants [74]. The increase in arabidopsis tolerance to salinity under the influence of *B. cereus* KP120 bacterium isolated from *K. virginica* was accompanied by the accumulation of proline in tissues [73].

In wheat plants, the endophytic strain *B. cereus* LN714048, isolated from buffel grass *Cenchrus ciliaris* L., caused an increase in salt tolerance under salinity stress, accompanied by an increase in the level of proline, phytohormones, antioxidant enzyme activity and, subsequently, an improvement of yield parameters such as the weight of seeds and ear length [80]. It was shown that the stimulation of *Cucumis sativus* L. seedling growth under the influence of *B. cereus* [137] and *Glycyrrhiza uralensis* Fisch. ex DC., 1825 under the influence of *B. cereus* G2 [138] under salt stress is associated with the accumulation of antioxidants. In the last case, bacterial treatment increases the content of proline and glycinbetaine, and the transcriptional activity of genes encoding betaine aldehyde dehydrogenase, α-glucosidase, and SS-genes, which regulate osmotic potential of plant cells.

*Bt* MH161336 treatment increased growth parameters in salinity stressed lettuce and caused the up-regulation of proline, superoxide dismutase, catalase, polyphenol oxidase, and peroxidase activity in plants [138]. Climate changes demand the investigation of the possibility of using endophytic and rhizospheric bacteria to promote plant tolerance to arid conditions of growth [139,140]. Thus, *Bt*AZP2 isolated from the roots of *Pinus ponderosa* trees, which grew under violent conditions (drought, nutrient limitation heat, and UV-irradiation stress) exhibited the high potential to enhance drought stress tolerance in wheat [61].

The treatment of soybean plants with *B. cereus* UFGRB2 and combined treatment with *B. cereus* CP003187.1 and *P. fluorescens* GU198110.1 influenced the efficiency of photosynthesis, maintaining the potential quantum yield of PSII and the rate of photosynthesis under the drought conditions, while these indicators decreased in non-inoculated plants [141]. *B. cereus* MKA4 treatment led to the activation of enzymes of the pro-/antioxidant system of drought-sensitive wheat variety HD2733 under stressful drought conditions [142]. The increase in tolerance to high temperature in soybean plants after their treatment with an endophytic strain of *B. cereus* SA1 was found. Thus, in *Bt* SA1-inoculated plants GmLAX3 and GmAKT2 genes were overexpressed, reactive oxygen species generation was decreased, which can be critical in plants under heat stress [81]. So, to summarize, through all these different abiotic influences, we think that the most important impact of *Bt* and *B. cereus* on stressed plants include its ability to decrease oxidative damage in plant cells.

Data on the ability of endophytic *B. cereus* strains to scavenge the air from ozone [143], formaldehyde [144], and ethylbenzene [145] in the plant–microorganism biome are of interest. A characteristic feature of the strain *Bt* ZS-19 isolated from the sludge of the wastewater containing pyrethroids is the ability to degrade 3-phenoxybenzoic acid and also a number of pyrethroid compounds [146]. In a recent study, *Bt* MB497, isolated from wheat rhizosphere in Pakistan, showed the ability to degrade up to 90.57% of 3,5,6-trichloro-2-pyridinol (pesticide chlorpyrifos) within 72 h in vitro [147]. It has been reported that the *Bt* strain from the commercial bioinsecticide Bac-Control WP from the company VectorControl (Brazil) is able to degrade the insecticide cypermethrin [148]. On the one hand, the last mentioned property opens new prospects for the disposal of dangerous pesticides, and on the other hand, this possibility shall be taken into consideration as a determining factor in plant protection systems, including biocontrol agents and chemical means.

The ability of strains of endophytic bacteria, including *Bt* and *B. cereus*, to detoxify pollutants opens up new opportunities for improving the ecological component of urbanized areas by forming “green” plant–microbial communities that can reduce the negative background of atmospheric and soil pollutants.

#### 2.2.5. Plant-Growth Promoting Effect

An endophytic lifestyle and the ability to protect plants from biotic and abiotic influences suggest that strains of *Bt* and *B. cereus* can improve viability and adaptive potential of the whole plant/microbes system, which leads to the increase in plant growth and yield [1,2,3,4,149,150]. It was assumed that the plant growth-stimulating effect of *Bt* and *B. cereus* strains is associated with the production of metabolites by bacteria, among which phytohormone-like compounds are the most discussed [151]. At the same time, surfactins, bacteriocins, and siderophores can also play a significant role in the process of inducing plant growth characteristics.

Plant-growth stimulation by *B. cereus* was documented, for example, for wheat, potato, pea, soybean, Chinese cabbage (*B. rapa* L. Chinensis group), maize, and rice plants [152,153,154,155]. For example, after the pre-sowing inoculation of rice with the *B. cereus* GGBSU-1 strain, the rate of their germination was increased up to 100% compared to 65% in controls, which was accompanied by corresponding changes in phenotypic (morphological parameters and the biomass of seedlings) and biochemical (the content of chlorophylls a and c, total soluble sugar, and α-amylase activity) parameters in seedlings [155]. The inoculation of plants with *B. cereus* T4S, isolated from the endosphere of *H. annuus* L. roots, promoted crop growth, including taproot length, root length, number and weight, and seed, weight compared to the controls [83]. The treatment of chickpea seeds with *B. cereus* MEN8 stimulated seed germination and plant growth [156]. The solubilization of phosphates and the formation of auxins, HCN, and ammonia by *B. cereus* LPR2 (isolated from the spinach rhizosphere) [152] and *Bt* KVS25 [157] are referred to as the backbone of the ability of these strains to stimulate the germination and growth of maize plants and the growth of *Brassica juncea* L., respectively.

The endophytic bacterium *Bt* W65, isolated from potato shoots, increased the duration of the flowering of plants of the corresponding culture by 8–13 days compared to control. The yield of tubers also increased by 7.9–14.6%, and with the joint inoculation of tubers with *B. amyloliquefaciens* and *Bt* W65 cells, an increase in the yield of large tubers and a decrease in the presence of infectious agents in plantings was observed [64]. Similarly, the potassium-solubilizing *B. cereus* strain WR34 efficiently promoted plant growth and the accumulation of dry biomass of the aerial part of the potato, which increased the yield of this crop by 20% compared to the control plots [153]. Tsvetkova P. et al. [158] showed that treatment of potato plantings with *Bt* ssp. *darmstadiensis Bt*H10 (RCAM 01490) stimulated the accumulation in the crop by up to 1.6 times and led to a more than three-fold decrease in the population of *L. decemlineata* and an up to twelve-fold decrease in the development of leaf and stolon rot caused by the fungus *Rhizoctonia* sp.

The seed treatment of spring rapeseed and wheat with *Bt* ssp. *fukuokaensis*; ssp. *toumanoffi*; ssp. *morrisoni*; ssp. *amagiensis*; and ssp. *dakota* led to a positive effect on the length of roots by up to 3.4 times and seedlings by up to 1.9 times, and a decrease in the number of seedlings affected by root rot caused by pathogenic fungi *F. oxysporum* and *A. alternata* [159]. The studied strains made it possible to obtain a higher quality and higher yield compared to the control variant. When using strains of *Bt* spp. *morrisoni* and *Bt* spp. *dacota* potato yield increased by 1.4 and 1.5 times, respectively [160].

The growth potential of *Lavandula dentata* L. was found to be preserved, even under drought conditions, after inoculation with a composition of five strains of arbuscular mycorrhizal fungi with an endophytic *Bt* isolate, which the authors attribute to the ability of bacteria and fungi to produce auxins and ACC deaminase and effectively immobilize phosphates [161]. A consortium containing strains of *B. cereus* LN714048 and *Pseudomonas moraviensis* LN714047 stimulated plant biometric parameters, including height, weight, and seed weight [162]. The complex use of *B. cereus* TSH77 and *B. endophyticus* TSH42 contributed to an increase in the biomass of turmeric (*Curcuma longa* L.) rhizomes and, accordingly, the content of curcumin [163]. The double inoculation of *Stevia rebaudiana* Bertoni, 1905 seedlings with cells of *B. cereus* SrAM1 and *A. brasilense* contributed to an increase in chlorophyll content, as well as the activity of antioxidant enzymes and the positive regulation of genes responsible for the biosynthesis of steviol glycoside [49,164].

## 3. Genetic Engineering Approach to Develop Next-Generation of *Bt*-Based Agents

Now, genetic modifications of *Bt* serve the purpose of dissolving two main problems of *Bt*-preparations, such as increase in bacteria tolerance to the impact of environmental factors and improving its insecticidal effects against different pests. The strategies of genetic engineering approaches to constructing strains with the required properties are as follows: (1) the up-regulation of the key enzyme gene involved in the target compound biosynthesis; (2) relieving the inhibition and/or repression of the key enzyme; and (3) the interruption of the pathways for synthesizing by-products [165]. The development of next-generation artificially improved *Bt* strains or strains heterologically producing *Bt*-toxins involves a broad spectrum of DNA reorganization, such as site-directed mutagenesis (SDM), the suppression and overexpression of genes, including RNA interference (RNAi) [166,167] Initially, RNAi was proposed as a suggestive strategy for the inhibition of viral infection. It is a post-transcriptional gene regulation mechanism characteristic of (possibly) all eukaryotes, including insect pests [106,107,108]. The mechanism is triggered by double-stranded RNA (dsRNA) precursors that are processed into short-interfering RNA (siRNA) duplexes, which then realize the recognition and repression of complementary dsRNAs, such as mRNAs or viral genomic RNAs [168].

### 3.1. Improvement of Insecticidal Properties of Bacterial Strains

Currently, genetic engineering approaches make it possible to transfer/supplement/modify genes encoding insectotoxin to other *Bt* strains or strains of another bacterial species using homologous recombination [169]. Current information on Cry- and non-Cry genes, which were used for the recombination of a broad spectrum of bacterial strains is assumed in [106,170]. An important tool for the recombination of *Bt* strains is site-specific recombination (SSR), which is useful for engineering strains with original combinations of Cry toxins genes with improved insecticidal activity [166,167,171]. It seems interesting to create endophytic *Bt* or other bacterial species whose populations in the internal tissues of plants would be safe from the environment and have greater activity against pests. These investigations originated in the last decade of the 20th century when the *Bt* gene encoding Cry1Aa was expressed in root-associated *P. fluorescens* [172]. Thus, the introduction of the cry1Ia gene into the endophytic strain *B. subtilis* 26D does not lead to the loss of the endophytic status of *B. subtilis* 26DCryChS line and gives impetus to its insecticidal and aphicidal activity in vitro and in planta [29,30]. Endophytic *Burkholderia pyrrocinia* JKSH007 heterologically expressing the *Btcry218* gene showed an effectiveness against *Bombyx mori* L. [173]. The ability of *Pantoea agglomerans* 33.1:pJTT expressing *cry1Ac7* to inhabit *Saccharum officinarum* L. tissues was confirmed by re-isolation from the plant’s rhizosphere, roots and shoots. Thus, the introduction of an exogenous gene did not affect the plant–host interaction but increased the mortality of *Diatraea saccharalis* Fabricius, 1794 fed on inoculated stems [174]. The transfer of “useful” insectotoxin genes from other economically important *Bt* strains to endophytic bacteria, as well as the maintenance of their consortiums, should contribute to the creation of new-generation biological agents based on them. At the same time, modern technologies for editing microbial genomes based on the CRISPRCas9 platform [168,175] can be proposed to disable the α-exotoxin and β-exotoxin synthesis of *Bt*.

### 3.2. Approaches to the Development of UV-Tolerant Bt

The problem of the UV-irradiation susceptibility of *Bt* seriously restricts its effective use. The exogenous addition of UV protective agents, such as rhodamine B or methyl green, can protect spores from the light [176]. Subsequently, for the same purpose, latex particles, ethanol, and olive oil have been used to encapsulate *Bt* in colloidosomes [177,178].

Homologous recombination technology was used for the insertion of the *yhfS* gene encoding acetyl-CoA acyltransferase in *Bt* LLP29 R-yhfS. The loss of the yhfS gene in the knockout strain *Bt* LLP29 Δ-yhfS led to the reduction in antioxidant ability and reduced UV resistance of the mutant [179]. The cell-surface exposure of chitinase Chi9602ΔSP was developed on the basis of *Bt* BMB171 using two repeat N-terminal regions of autolysin (Mbgn)2 as the anchoring motif. After continuous culturing for 120 h, the line of *Bt* expressing chitinase Chi9602ΔSP showed narrow pH tolerance and obviously enhanced UV radiation resistance capacity in addition to a high inhibitory effect towards phytopathogenic fungi, *F. oxysporum* FB012 and *Botryosphaeria berengeriana* FB016 [180]. CRISPR/Cas9 systems have been used to knock out the homogentisate-1,2-dioxygenase (*hmgA*) gene and obtain a melanin-producing mutant *Bt* HD-1-1 hmgA. The anti-UV test shows that melanin arranges protection to both *Bt* cells and Cry toxin crystals. After UV-irradiation the strain *Bt* HD-1-1 hmgA still had an 80% insecticidal activity against *H. armigera*, while the wild line only had about 20% [8].

Cry genes have been expressed in *P. fluorescens* and *Anabaena* sp. to increase the damage to crystals from UV light [181,182], as well as in *E. coli*; *B. megaterium* [183]; *B. subtilis* [83]; *Clavibacter xyli* Davis et al. 1984; *Herbaspirillum seropedicae* Baldani et al. 1986; *R. leguminosarum* [170]; *Beauveria bassiana* (Bals.-Criv.) Vuill., 1912 [184]; etc. The recombinant strain *P. fluorescens* is the base of biopesticide “CellCapTM” (Mycogen Corp.; Indianapolis, IN, USA), contains encapsulated Cry toxins [185]. The increase in the amount of *Bt*-plants can be partially attributed to the means of the protection of insectotoxins from UV rays [72].

### 3.3. Bt Crops Prospects

Since 1996, genetically engineered *Bt* crops have been planted in the fields, which led to a “gene revolution” in agricultural production [186,187]. By the early 21th century, *Bt*-potato, *Bt*-cotton, *Bt*-maize, *Bt*-eggplant, etc. were actively distributed worldwide, which allowed for a significant reduction in the amount of chemical insecticides used in a number of countries [96,188]. However, this approach lead to a fairly rapid spread of resistant pest populations [80,189,190]. To overcome insect resistance, it is possible to introduce *Bt* crops containing more than two genes encoding insecticidal proteins [189,190]. The Bollgard cotton variety bearing *Cry1Ac* gene decreased the viability of pink bollworm *Pectinophora gossypiella* (Saunders, 1844) and corn earworm *Helicoverpa zea* Boddie, 1850. Plants of the Bollgard II variety, expressing two *Bt* endotoxins, expand the spectrum of protective features against lepidopteran pests [191]. *Bt*-cotton with cassettes of protective genes (1Ac/Cry2Ab/Vip3A), (Cry1Ab/Cry2Ac/Vip3Aa19) or (Cry1Ac/Cry1F/Vip3A) was cultivated in the 2016–2017 season on more than 90% of the arable lands of Australia [192].

Currently, the creation of plants containing not only *Cry* or *Vip* genes but also containing other gene sequences is of interest in order to increase the effectiveness of biological plant protection against pests. Recently, the US EPA approved a transgenic corn, SmartStaxPRO, expressing the Cry3Bb1 protein, and a dsRNA complementing the RNA of the vacuolar protein DvSnf7 of *Diabrotica virgifera* LeConte, 1858 [193,194]. A vector containing information about dsRNA targeting the acid methyltransferase gene of the juvenile hormone biosynthesis of *H. armigera* was inserted into the genome of *Bt*-cotton and impaired the resistance of pest compared to plants expressing only insectotoxic proteins [195]. The RNAi-mediated knockdown of *H. armigera* acetylcholinesterase, the ecdysone receptor, and v-ATPase-A genes by producing dsRNAs homologous to genetic targets in potato plants led to mortality and abnormal development in the larva of this insect (recorded ten days post feeding) [167].

Apparently, the application of single *Bt* genes to modify plant genomes will be gradually replaced by multiple *Bt* toxin genes or *Bt* with other nucleotide sequences.

### 3.4. Bt as a Means of dsRNAs Deliverance

The important problem of interference methods is dsRNA degradation by nucleases in the gut lumen and tissues of insects. Retaining dsRNA molecules in the gut or hemocoel of pest insects is the key aspect of an effective dsRNA delivery [166]. Thus, dsRNase catalyzing the specific cleavage of dsRNA has been found in the saliva of *Lygus lineolaris* Palisot de Beauvois, 1818 [196]; pea aphid *Acyrthosiphon pisum* Harris, 1776 [197]; *Schistocerca gregaria* Forskal, 1775 [198]; *H. armigera* [167]; etc. A dsRNA cassette targeting the multiple genes of *H. armigera* revealed more rapid cleavage in midgut juice compared to the hemolymph [167], and for this reason, the *Bt*-mediated appearance of pores in digestion membranes, in our opinion, can improve the efficacy of dsRNAs along with dsRNAs targeting dsRNases [199]. The stability of mRNA provided by, for example, the Shine–Dalgarno sequence (GAAAG-GAGG), is a promising factor of the high-level expression of Cry genes in *Bt* [167,200,201]. The binding of the 30S ribosomal subunit to this sequence might prevent mRNA cleavage by RNAses of pest. The use of a sporulation-dependent promoter of Cry genes of *Bt* for the transcription of the target dsRNA sequence, leads to the fact that the dsRNA will be spontaneously produced during the sporulation phase [201,202]. It has been demonstrated that the incorporation of plasmid pBtdsSBV-VP1, which carries out dsRNA complemental to the VP1 sequence of the sacbrood virus (SBV) in *Bt* 4Q7 and the subsequent appliance of exogenous total RNA, leads to a decrease in SVB severance in *Apis cerana* (Fabricius, 1793) families [202]. Then, the plasmid p*Bt*dsSBV-VP1 was inserted into the *Bt* NT0423, which expresses Cry1 protein, resulting in SBV replication being repressed in *A. cerana* bees as well as the viability of the *A. cerana* parasite *Galleria mellonella* L. [200]. These results demonstrated that dsRNA-expressing *Bt* products could be efficiently exploited for the control of both viral diseases and insect pests simultaneously.

And furthermore, it is possible to enhance the toxicity of Cry toxins using dsRNA cassettes. Thus, *Bt* strains 8010AKi and BMB171AKi expressing the dsRNA of the arginine kinase gene (PxAK) of *P. xylostella*, flanking two ends with the promoter Pro3α, effectively decreased PxAK expression in ones treated with the composition with wild Cry-producing *Bt* 8010 and caused a higher lewel of mortality of the pest [203]. Separately, *E. coli* HT115 dsINT expressing the dsRNA of integrin β1 subunit gene (SeINT) cause a less than 50% mortality rate against *Spodoptera exigua* Hubner, 1808 larvae, and *E. coli* expressing Cry1Ca led to a maximal 58% mortality rate of the pest. When *S. exigua* larvae were treated with the Cry1Ca-expressing bacteria (*E. coli* or *Bt* subsp. *aizawai* from commercial Xentari insecticide) after treatment with *E. coli* HT115 dsINT, the insecticidal activity of the Cry1Ca was significantly enhanced up to about 80% [201]. The nuclease gene HaREase characteristic for Lepidoptera is up-regulated by dsRNA and affects RNAi in *H. armigera*. When this gene was knocked out using the CRISPR/Cas9 system, the midgut epithelium structure was not affected in the ΔHaREase mutant, but when larvae were fed an artificial diet with sublethal doses (2.5 or 4 μg/g) of Cry1Ac, the growth rate of the ΔHaREase line was repressed significantly [204]. The insecticidal activity of the *Bt*-based biopesticide Xentari™ (Valent BioSciences) against larvae of *S. littoralis* was significantly enhanced by pre-treatment with dsRNA-Bac targeted against the *Sl* 102 gene, which is responsible for insect cell aggregation and encapsulation to protect against *Bt* infection [205]. Likewise, the efficacy of biological preparations based on live *Bt* cells was enhanced when used together with dsRNA-Bac specific to sequences of the *P. xylostella* Pxf gene [206], which caused insect resistance to Cry1Ac toxin. The RNAi-mediated suppression of the Cat L-like gene encoding the lysosomal cathepsin L-like cysteine protease of *Bombyx mori* led to an increase in larvae mortality under the influence of *Bt* subsp. *kurstaki* strain ABTS-351 (Dipel^®^, Valent BioSciences, Libertyville, IL, USA) [207]. It is probably that the increase in insect resistance to biocontrol agents based on *Bt* strains producing toxins and the use of this bacterium as a platform for the expression of dsRNA can help in pest control using the Cry + RNAi strategy [205,206,207].

## 4. Conclusions

Currently, a broad spectrum of data has been accumulated on the influence of *Bt* and *B. cereus* on different species of plants. New studies in the field of interactions between plants and bacteria reveal the ability of *Bt* and *B. cereus* to invade and exist in plant microbiomes, where bacteria possess protection against environmental factors, in particular, UV radiation. New algorithms, which can be called “microbiome engineering” can detect, modulate, and enhance benefits and ways to improve the efficacy of strains. Unfortunately, our knowledge on the intricate interactions of plants with beneficial microbes is quite limited.

It is clear that the colonization of plant niches by microorganisms, including *Bt* and *B. cereus*, is a multifaceted process consisting of different steps that are mediated by plant and/or bacterial molecular patterns [208]. These interactions can be strictly specific for different plant and microbe species and can be dependent on environmental conditions. Now, there are no data on universe genes or molecular patterns, which determine the plant-associated status of microbial strains [83,208]. Therefore, the plant should be considered as an integral phytoholobiont, wherein the artificial introduction of endophytes which makes it possible to effectively protect the macroorganism not only from attacks by phytophagous insects, phytopathogenic viruses, bacteria, and fungi but also from unfavorable abiotic environmental factors, as well as pollutants. This approach makes it possible to obtain products that are safe for humans and animals in cenoses, which cannot avoid urban influences. The identification of additional abilities of endophytes to metabolize hazardous chemical compounds and increase the resistance of plants to heavy metals makes it possible to effectively use the above and other similar bacteria also for the phytoremediation of territories contaminated with various pollutants, including the air environment, which is becoming increasingly important for humankind around the world.

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
