# Peer review of "Plant-Associated Bacillus thuringiensis and Bacillus cereus: Inside Agents for Biocontrol and Genetic Recombination in Phytomicrobiome"

_plants, 2023, doi:10.3390/plants12234037_

Round 1
Reviewer 1 Report
Comments and Suggestions for Authors
In this review, the author summarizes current data on the mutualistic relationships of Bt and B. cereus with plants, considering these microorganisms not only the least environmentally hazardous insecticides, but also unique tools of genetic engineering and biological control, based on the mechanisms of RNA-interference. The ablility of Bt to invade and exist in plant microbiomes makes it possible to artificial introduction of endophytes into plants which could protect the macroorganism not only from attacks by phytophagous insects, phytopathogenic viruses, bacteria, and fungi, but also from unfavorable abiotic environmental factors, as well as pollutants. However, even though Bt and B. cereus are closely related genetically, as the author mentioned in introduction that strains of these two species are not distinguishable using some genetic studies such as MLST, it's not appropriate to consider the Bt and B. cereus as a single species yet. Given that B. cereus is an opportunistic pathogen capable of causing foodborne illnesses in humans, distinguishing between the safe biopesticide Bt and the human pathogenic B. cereus has always been crucial for registering Bt biological control agents with a Generally Regarded as Safe (GRAS) status. And in recent years, many studies have attempted to address this issue using genome-based taxonomic frameworks, and propose simple means to characterize new Bt strains from other closely related B. cereus species [1-4]. The author should consider these studies and cite them in this paper. Given that the review was mainly discussed from the perspective of Bt, we suggest that the author focuses only on Bt as a potential inside agent for biocontrol and genetic recombination in phytomicrobiome in this review.
Totally, they did not discuss much about the classification of Bt and Bc in the article, nor did they cite several very crucial articles including some papers that used genomic methods to classify, but according to their writing, they considered Bt and Bc as a same species, and they did not seem to care much about the classification of Bt and Bc. But this review as a whole is basically from the Bt perspective, so it may be better for them to write only from the Bt perspective.
Major revision.
Some important papers recommended to you, could be cited in your manuscript:
1. Baek, I.; Lee, K.; Goodfellow, M.; Chun, J. Comparative genomic and phylogenomic analyses clarify relationships within and between Bacillus cereus and Bacillus thuringiensis: Proposal for the recognition of two Bacillus thuringiensis genomovars. Front Microbiol. 2019, 10, 1978.
2. Carroll, L.M.; Wiedmann, M.; Kovac, J. Proposal of a taxonomic nomenclature for the Bacillus cereus group which reconciles genomic definitions of bacterial species with clinical and industrial phenotypes. mBio 2020, 11, e00034–e00020.
3. Carroll, L.M.; Cheng, R.A.; Kovac, J. No assembly required: Using BTyper3 to assess the congruency of a proposed taxonomic framework for the Bacillus cereus group with historical typing methods. Front. Microbiol. 2020, 11, 580691.
4. Wang K, Shu C, Bravo A, Soberón M, Zhang H, Crickmore N, Zhang J. Development of an Online Genome Sequence Comparison Resource for,Bacillus cereus sensu lato Strains Using the Efficient Composition Vector Method. Toxins (Basel). 2023 Jun 12;15(6):393.

Comments on the Quality of English Language
Minor editing of English language required
Author Response
We would like to express our gratitude for all that the reviewer has done. Please, find our answers below.
- In this review, the author summarizes current data on the mutualistic relationships of Bt and B. cereus with plants, considering these microorganisms not only the least environmentally hazardous insecticides, but also unique tools of genetic engineering and biological control, based on the mechanisms of RNA-interference. The ablility of Bt to invade and exist in plant microbiomes makes it possible to artificial introduction of endophytes into plants which could protect the macroorganism not only from attacks by phytophagous insects, phytopathogenic viruses, bacteria, and fungi, but also from unfavorable abiotic environmental factors, as well as pollutants. However, even though Bt and B. cereus are closely related genetically, as the author mentioned in introduction that strains of these two species are not distinguishable using some genetic studies such as MLST, it's not appropriate to consider the Bt and B. cereus as a single species yet. Given that B. cereus is an opportunistic pathogen capable of causing foodborne illnesses in humans, distinguishing between the safe biopesticide Bt and the human pathogenic B. cereus has always been crucial for registering Bt biological control agents with a Generally Regarded as Safe (GRAS) status. And in recent years, many studies have attempted to address this issue using genome-based taxonomic frameworks, and propose simple means to characterize new Bt strains from other closely related B. cereus species [1-4]. The author should consider these studies and cite them in this paper. Given that the review was mainly discussed from the perspective of Bt, we suggest that the author focuses only on Bt as a potential inside agent for biocontrol and genetic recombination in phytomicrobiome in this review.
We are sorry that we didn’t pay enough attention to these important topics in the review. Problems mentioned above were discussed in the revised paragraph below. We would like to thanks reviewer for very reasonable references, that helped us to elucidate this topic more objectively. Besides, through the whole text we marked the difference between B. cereus sensu lato and sensu stricto (if possible).
In the text:
Bt is accepted by many authors as a single species of B. cereus group, but from different points of view Bt can be seen as an independent species or a subspecies of B. cereus sensu lato, bearing plasmids encoding insectotoxic proteins [10; 11]. According to several experts, Bt are insectotoxin producers and belong to the supraspecific group of B. cereus sensu lato, which includes 21 closely related species, including Bt, B. mycoides Flügge 1886, B. weihenstephanensis Lechner et al. 1998, B. pseudomycoides Nakamura 1998, B. anthracis Cohn 1872, B. cereus sensu stricto, B. gaemokensis Jung et al. 2010, B. manliponensis, B. toyonensis, B. bingmayongensis, B. cytotoxicus, and B. wiedmannii [12]. B. cereus sensu stricto (Bacteria; Terrabacteria group; Bacillota; Bacilli; Bacillales; Bacillaceae; Bacillus; Bacillus cereus group) toxins causing gastrointestinal diseases, wound or systemic infections, and eye infections [11]. Perhaps because of the controversies surrounding the approach a lot of molecular investigations have shown that strains of these species are not distinguishable, since DNA–DNA hybridization similarities and average nucleotide identity between these Bacillus types are a little higher than the levels used to distinguish between closely related species, but B. cereus sensu stricto and Bt should continue to be recognized as validly published species [13, 14]. The extensive analysis of genomic data show that the distribution of insecticidal genes is irregular and numerous strains identified as Bt can be assigned to polyphyletic subclades within the B. cereus/Bt clade [14]. Thus, presence of certain plasmid encoding Cry and Vip cannot be used as a diagnostic marker of Bt. According to the composition vector tree (CVTree) method there is the relationship between Bt and other B. cereus sensu lato species but it was the best option to be used for typing Bt strains [15]. .On the basis of average nucleotide identity researchers set up Bt (so-called by authors B. cereus sensu stricto serovar Berliner biovar Thuringiensis) in B. cereus sensu stricto genomospecies containing 949 genomes [16]. Proximity of Bt to pathogenic microorganisms, for example to B. cereus, put forward the problem of their identification and separation in food products [17, 18]. Now, Bt residues which cannot be distinguished from natural populations of B. cereus in routine food safety diagnostics present are enumerated as “presumptive B. cereus” [17] and approaches to its differentiation often based on Cry-genes presence [18]. Thus, information on Bt and B. cereus features, set out in this review, based on the definition of strains and isolates proposed by researchers, but it should be noted that some of them can be attributed to each other. Analysis of the above sources allows us to conclude that it is necessary to draw attention to the problem of B. cereus group identification using various taxonomic methods.
- Totally, they did not discuss much about the classification of Bt and Bc in the article, nor did they cite several very crucial articles including some papers that used genomic methods to classify, but according to their writing, they considered Bt and Bc as a same species, and they did not seem to care much about the classification of Bt and Bc. But this review as a whole is basically from the Bt perspective, so it may be better for them to write only from the Bt perspective.
We agree that classification of the B. cereus group now is the subject of sharp debate. We are sorry that we might have given the impression that the review is based only on Bt features and perspectives. We have tried to highlight that new studies in the field of interaction between plants and bacteria reveal the ability of both Bt and B. cereus to invade and exist in plant microbiomes and bring benefits to plants. Unfortunately, our knowledge on B. cereus is quite limited, but, in our opinion, it is of great importance to discuss Bt and B. cereus to distinguish their possibilities and place in agrocenoses. Clarifications have been made to the text of the article.

Reviewer 2 Report
Comments and Suggestions for Authors
No comments.
Comments on the Quality of English Language
No comments.
Author Response
We would like to express our gratitude for all that the reviewer has done. Moderate editing of English language was carryed out.

Reviewer 3 Report
Comments and Suggestions for Authors
Overall, the manuscript is well-organized and written in a clear manner. However, in some sections, the writing could be further refined to enhance clarity. We recommend proofreading the manuscript thoroughly to correct any typographical errors and to ensure a consistent writing style.
- The abstract provides a concise summary of the study's objectives, methods, and findings. Summarize the key findings and implications of the review in a clear and succinct manner.
- Mention the specific aspects of biocontrol and genetic recombination
- However, I suggest including specific findings in the abstract to make it more informative and enticing for potential readers.
- Provide a concise background on the significance of plant-associated microorganisms in agriculture and environmental sustainability. However, it would be beneficial to expand on the significance and relevance of exploring the Bacillus thuringiensis and Bacillus cereus: inside agents for biocontrol, and how it could potentially contribute to the understanding of improve crops productivity or the broader field of genetics. Provide a comprehensive overview of these bacterial species, including their taxonomy, morphology, and natural habitats if possible.
- Consider specifying the focus and importance of these bacteria in the phytomicrobiome.
- Emphasize their relevance in plant-microbe interactions.
- Section 2.1 Detail the various mechanisms employed by Bacillus thuringiensis and Bacillus cereus in biocontrol, such as toxin production, competition, and induced systemic resistance.
- Describe the intricate interactions between these Bacillus species and their host plants, highlighting any co-evolutionary patterns or specific plant responses.
Author Response
We would like to express our gratitude for all that the reviewer has done. Please, find our answers below.
Overall, the manuscript is well-organized and written in a clear manner. However, in some sections, the writing could be further refined to enhance clarity. We recommend proofreading the manuscript thoroughly to correct any typographical errors and to ensure a consistent writing style.
- The abstract provides a concise summary of the study's objectives, methods, and findings. Summarize the key findings and implications of the review in a clear and succinct manner.
Mention the specific aspects of biocontrol and genetic recombination. However, I suggest including specific findings in the abstract to make it more informative and enticing for potential readers.
The abstract was strictly revised:
Abstract: Bacillus thuringiensis Berliner (Bt) and B. cereus sensu stricto Frankland and Frankland are closely related species of aerobic, spore-forming bacteria included in B. cereus sensu lato group. This group is one of the most studied, but it remains also the most mysterious species of bacteria. Despite more than a century of research on the features of these ubiquitous bacteria, there are a lot of questionable issues related to their taxonomy, resistance to external influences, endophytic existence, their place in multidimensional relationships in the ecosystem, and many others. The review summarizes current data on the mutualistic relationships of Bt and B. cereus bacteria with plants, structure of the phytomicrobiomes including Bt and B. cereus, and abilities of plant-associated and endophytic strains to improve plant resistance to various environmental factors and its productivity. Key findings on the possibility of the use of Cry gene promoter for transcription of the target dsRNA and simultaneous release of pore-forming proteins and provocation of RNA-interference in pest organisms allow us to consider this group of microorganisms as unique tools of genetic engineering and biological control. This will open the prospects for the development and direct change of plant microbiomes, and possibly serve as the basis for the regulation of the entire agroecosystem.
- Provide a concise background on the significance of plant-associated microorganisms in agriculture and environmental sustainability.
It was specified:
Now the niche of these bacteria is more often attributed to the phylloplane, considering them to be mutualists in relation to plants [24] and, in our opinion, it is a kind of revolution in the concept of ecosystem. Plants intimately interact with diverse communities of microorganisms, such as bacteria, fungi, nematodes, protists and viruses that colonize all plant tissues, rhizosphere and soil [1, 19, 23, 25]. The microbiome establishes complex and dynamic interactions with the host plant and can improve plant resilience to environmental stresses due to the high level of flexibility of these important genetic resources of the whole plant/microbiome system [25, 26].
- However, it would be beneficial to expand on the significance and relevance of exploring the Bacillus thuringiensis and Bacillus cereus: inside agents for biocontrol, and how it could potentially contribute to the understanding of improve crops productivity or the broader field of genetics.
The structure of the review was improved to highlight the significance of plant-associated Bt and B. cereus.
- Provide a comprehensive overview of these bacterial species, including their taxonomy, morphology, and natural habitats if possible.
These parameters were described (“Rod-shaped, poorly motile, spore-forming, facultatively anaerobic, synthesizing insectotoxic proteins Bt (Bacteria; Terrabacteria group; Bacillota; Bacilli; Bacillales; Bacillaceae; Bacillus; Bacillus cereus group) subspecies thuringiensis, kurstaki, aizawai, tenebrionis, and israelensis are most often used as the basis of bioinsecticides worldwide [1, 4].” and “B. cereus sensu stricto (Bacteria; Terrabacteria group; Bacillota; Bacilli; Bacillales; Bacillaceae; Bacillus; Bacillus cereus group)”).
Taxonomy of species under consideration was clarified.
- Consider specifying the focus and importance of these bacteria in the phytomicrobiome.
Emphasize their relevance in plant-microbe interactions. Describe the intricate interactions between these Bacillus species and their host plants, highlighting any co-evolutionary patterns or specific plant responses.
Unfortunately, we must recognize that there isn’t enough data on the role of this group of bacteria in microbiome (but by and large, there is just a little data on the role of any other group of bacteria in microbime and functioning of microbiome as whole). We specifyed these future perspectives:
New algorithms, which can be called “microbiome engineering” can detect, modulate and enhance benefits and ways to improve the efficacy of strains. Unfortunately, our knowledge on intricate interactions of plants with beneficial microbes is quite limited.
It is clear that colonization of plant niсhes by microorganisms, including Bt and B. cereus, is a multifaceted experience consisting different stadiums that are mediated by plant and/or bacterial molecular patterns [208]. These interactions can be strictly specific for different plant and microbe species and can be depend on environmental conditions. Now there isn’t any data on universe genes or molecular patterns, which determine plant-associated status of microbial strain [83, 208]. Therefore, the plant should be considered as an integral phytoholobiont, the artificial introduction of endophytes into which makes it possible to effectively protect the macroorganism not only from attacks by phytophagous insects, phytopathogenic viruses, bacteria, and fungi, but also from unfavorable abiotic environmental factors, as well as pollutants.
- Section 2.1 Detail the various mechanisms employed by Bacillus thuringiensis and Bacillus cereus in biocontrol, such as toxin production, competition, and induced systemic resistance.
It was specifyed:
Now it is believed that systems of plants and its associated microbiota resulting from the evolutionary selection contributes to the overall stability of the whole holobiont [23, 33, 83]. Biocontrol of pathogens and pests by benefit microorganisms, originating from competition for niches/nutrients, possibly, one of the most important, at least now, features of microbiomes in agrocenosis [23-24]. Mechanisms of biocontrol on direct (production of antimicrobial/insecticide compounds) and indirect (induction of systemic resistance in plants) means can be conditionally divided [33]. These branches determine all spectrum of biocontrol possibilities of plant-associated microbes, including, for example, pathogen quorum sensing interference and altering of the soil microbiota [83].

Reviewer 4 Report
Comments and Suggestions for Authors
something to review in the systematics

Comments on the Quality of English Language
Minor editing of English language required
Author Response
We would like to express our gratitude for all that the reviewer has done. Сorrections made to the revised text are listed below:
Editing of English language was carried out.
Authority and systematics were added.
Spelling and italics of taxons were corrected.
